Azoramide ameliorates cadmium-induced cytotoxicity by inhibiting endoplasmic reticulum stress and suppressing oxidative stress

Zhang Lingmin 1
Zhang Jianguo 1
Zhou Yingying 1
Xia Qingqing 1
Xie Jing 1
Zhu Bihong 2
Wang Yang 3
Yang Zaixing 1 yangzaixingdiyi@163.com
Li Jie 1 liyijie12580@126.com
1 Department of Laboratory Medicine, Huangyan Hospital, Wenzhou Medical University , Taizhou, Zhejiang Province , People’s Republic of China
2 Department of Neurology, Huangyan Hospital, Wenzhou Medical University , Taizhou, Zhejiang Province , People’s Republic of China
3 Department of Gastroenterology, Shulan (Hangzhou) Hospital , Hangzhou, Zhejiang Province , People’s Republic of China
Bhardwaj Rajesh
Electronic publication date: 2024 Jan 31
Publication date: 2024
Volume: 12
Electronic Location ID: e16844
Received 2023 Jul 26; Accepted 2024 Jan 7
Copyright: © 2024 Zhang et al.
Copyright year: 2024
Copyright holder: Zhang et al.
License: This is an open access article distributed under the terms of the Creative Commons Attribution License, which permits unrestricted use, distribution, reproduction and adaptation in any medium and for any purpose provided that it is properly attributed. For attribution, the original author(s), title, publication source (PeerJ) and either DOI or URL of the article must be cited.
License URL: https://creativecommons.org/licenses/by/4.0/

Keywords: Azoramide, Cadmium, Endoplasmic reticulum stress, Oxidative stress, Inductively coupled plasma‒mass spectrometry

Funding: Medical Science and Technology Program of Zhejiang Province 2022RC083 and 2022RC295 Scientific Research Project of Taizhou Science and Technology Bureau in Zhejiang Province 20ywa39, 20ywa45, and 20ywb66 This study was supported by the Medical Science and Technology Program of Zhejiang Province (2022RC083 and 2022RC295) and the Scientific Research Project of Taizhou Science and Technology Bureau in Zhejiang Province (20ywa39, 20ywa45, and 20ywb66). The funders had no role in study design, data collection and analysis, decision to publish, or preparation of the manuscript.

==============================
Background

Cadmium (Cd) is hazardous to human health because of its cytotoxicity and long biological half-life. Azoramide is a small molecular agent that targets the endoplasmic reticulum (ER) and moderates the unfolded protein response. However, its role in Cd-induced cytotoxicity remains unclear. This study was performed to investigate the protective effect of azoramide against Cd-induced cytotoxicity and elucidate its underlying mechanisms.

Methods

Inductively coupled plasma‒mass spectrometry was used to measure Cd concentrations in each tissue of ICR male mice. The human proximal tubule epithelial cell line HK-2 and the human retinal pigment epithelial cell line ARPE-19 were used in the in vitro study. Cell apoptosis was determined by DAPI staining, JC-1 staining, and annexin V/propidium iodide double staining. Intracellular oxidative stress was detected by MitoSOX red staining, western blot, and quantitative real-time PCR. Moreover, ER stress signaling, MAPK cascades, and autophagy signaling were analyzed by western blot.

Results

The present data showed that Cd accumulated in various organs of ICR mice, and the concentrations of Cd in the studied organs, from high to low, were as follows: liver > kidney > testis > lung > spleen > eye. Our study demonstrated that azoramide inhibited ER stress by promoting BiP expression and suppressing the PERK-eIF2α-CHOP pathway. Additionally, we also found that azoramide significantly decreased ER stress-associated radical oxidative species production, attenuated p38 MAPK and JNK signaling, and inhibited autophagy, thus suppressing apoptosis in HK-2 and ARPE-19 cells.

Conclusion

Our study investigated the effect of azoramide on Cd-induced cytotoxicity and revealed that azoramide may be a therapeutic drug for Cd poisoning.

Introduction

Cadmium (Cd) is a heavy metal that may accumulate in the body by occupational exposure, uptake with diet, or inhalation through cigarette smoking (Filippini et al., 2020). Cd has considerable negative impacts on human health because it may cause a broad range of cytotoxic effects, especially to the liver, kidney, testicle, and retina (Komoike, Inamura & Matsuoka, 2012; Man et al., 2023; Zhang et al., 2019). A few studies have investigated the distribution of Cd in various organs and its accumulation profile (Egger et al., 2019; Tai et al., 2022). Chelating agents are commonly used to treat acute Cd poisoning. However, most chelating agents are associated with adverse reactions, which could potentially increase the Cd concentration in the kidney and worsen renal injury (Wu et al., 2004). Therefore, exploring and developing alternative therapeutic drugs or strategies is an urgent clinical need, and finding potential new drugs that possess novel mechanisms of action is the first priority.

Accumulating evidence has demonstrated that the endoplasmic reticulum (ER) is a major target organelle in Cd-exposed cells (Zhang et al., 2019). The ER, essential for intracellular homeostasis, plays a critical role in synthesizing, folding, modifying, and delivering biologically active proteins. ER dysfunction leads to the accumulation of unfolded proteins in the lumen and triggers the unfolded protein response (UPR). It is well-known that three ER transmembrane proteins serve as sensors of unfolded protein accumulation in the ER: PKR (RNA-activated protein kinase)–like endoplasmic reticulum kinase (PERK), activating transcription Factor 6 (ATF6), and inositol-requiring enzyme 1 (IRE1), which activate three branches of signal transduction to restore ER homeostasis (Komoike, Inamura & Matsuoka, 2012; Li et al., 2015; Zhang et al., 2020). However, the persistence of ER injury eventually results in ER stress and initiates programmed cell death (Ong & Logue, 2023; Zhang et al., 2022). These observations imply that attenuating ER stress and re-establishing ER homeostasis may be effective for the prevention of Cd-induced cytotoxicity.

Azoramide (N-(2-(2-(4-chlorophenyl)-4-thiazolyl) ethyl) butanamide) is a small molecular agent that targets the ER and modulates the UPR (Fu et al., 2015). Studies have suggested that azoramide can increase protein folding activity in the ER and promote chaperone expression. However, there have been limited studies on the pharmacological activities of azoramide. Previous studies have demonstrated that azoramide exhibits antidiabetic effects, alleviates nonalcoholic fatty liver disease, protects against dopaminergic neuron damage, attenuates high-frequency electrical stimulation-induced atrial myocyte injury, and enhances the differentiation of mesenchymal stem cells into fat cells (Bagci, Sahinturk & Sahin, 2019; Fu et al., 2015; Ke et al., 2020; Miao et al., 2022; Okatan et al., 2019; Ruan et al., 2018). However, whether azoramide can protect cells from Cd overload-induced cytotoxicity remains unclear.

Therefore, in the present study, we aimed to investigate the protective effect of azoramide against Cd-induced cytotoxicity and elucidate its underlying mechanisms. We also measured the Cd concentrations in the liver, kidney, spleen, lung, eye, and serum by inductively coupled plasma‒mass spectrometry (ICP‒MS).

Materials and Methods

Reagents, antibodies, and cell lines

Cadmium chloride (CdCl2) (purity > 99%) was purchased from Aladdin (C116344; Shanghai, China), prepared as a stock solution in ultrapure water and stored at −20 °C. Azoramide was obtained from Selleck (S8304; Shanghai, China), dissolved in dimethyl sulfoxide (D8418; Sigma−Aldrich, St Louis, MO, USA) as a stock solution and stored at −20 °C in the dark. 4′,6-Diamidino-2-phenylindole (DAPI) was purchased from Sigma−Aldrich (28718-90-3). An inhibitor of the p38 mitogen-activated protein kinase (MAPK) SB203580 (S1076) and a c-Jun N-terminal kinase (JNK) inhibitor SP600125 (S1460) were obtained from Selleck.

Primary antibodies against immunoglobulin heavy chain binding protein (BiP) (11587-1-AP; 1:1,000), phosphorylated JNK (p-JNK) (80024-1-RR; 1:1,000), phosphorylated p38 MAPK (p-p38 MAPK) (28796-1-AP; 1:1,000), phosphorylated α subunit of eukaryotic initiation Factor 2 (p-eIF2α) (28740-1-AP; 1:1,000), p-PERK (29546-1-AP; 1:1,000), nuclear factor-like 2 (Nrf2) (80593-1-RR; 1:1,000) and GAPDH (60004-1-Ig; 1:5,000) were obtained from Proteintech (Wuhan, Hubei, China). An antibody against light chain 3B (LC3B) (L7543; 1:1,000) was purchased from Sigma−Aldrich, and an antibody against poly ADP-ribose polymerase-1 (PARP1) (9532S; 1:1,000) was provided by Cell Signaling Technology (Danvers, MA, USA).

The human retinal pigment epithelial cell line ARPE-19 and the human proximal tubule epithelial cell line HK-2 were obtained from the National Collection of Authenticated Cell Cultures (Shanghai, China). Cells were maintained in DMEM/F12 medium (D0697; Sigma−Aldrich, St Louis, MO, USA) supplemented with 10% fetal bovine serum (FBS, S-FBS-SA-015; Serana, Pessin, Brandenburg, Germany) and 1% (v/v) penicillin/streptomycin (15070063; Gibco, Thermo Fisher Scientific, Waltham, MA, USA) in a humid atmosphere of 5% CO2 at 37 °C.

In vivo experiments

Specific pathogen-free male (SPF) ICR mice weighing 22 to 25 g (aged 5 weeks) (n = 10) were purchased from Beijing Vital River Laboratory Animal Technology Co., Ltd. All mice were housed in individually ventilated cages under SPF conditions in a controlled environment (24 °C, 55% humidity, and a 12-h day/night cycle) and given free access to drinking water and diet. All mice were kept in SPF facilities of Hangzhou Hibio Technology Company, and the experimental protocol was approved by the Animal Care and Use Committee of Hangzhou Hibio Technology Company (approval number HB2108002). The study was performed according to international, national and institutional rules for animal experiments.

The mice were allowed to acclimate for 1 week and were randomly divided into two groups. The concentrations of Cd were selected based on previous studies (Luo et al., 2016; Zhao et al., 2021). For cadmium exposure, ICR mice (n = 5) were intraperitoneally administered CdCl2 (5 mg/kg) once a day for 7 days. The normal control (n = 5) received the same volume of normal saline. Twenty-four hours after the last injection, all mice were anesthetized with pentobarbital sodium (70 mg/kg), and blood samples were collected for subsequent analyses. The mice were then sacrificed by cervical dislocation, and organs (heart, liver, kidney, lung, spleen, testis, and eyes) were dissected, weighed, and stored at −80 °C until analysis. No animal was excluded from analysis, but a mouse’s spleen weight, a mouse’s liver Cd concentration, and an animal’s serum Cd concentration in the control group were not included in the analysis because of the failure of sample collection or preparation. The organ coefficient (relative organ weight) was calculated with the organ-to-body weight ratio (%).

Measurement of the concentration of Cd in mouse blood and tissues

The concentrations of Cd in ICR mouse blood and tissues, including the liver, spleen, lung, kidney, testis, and eye were analyzed by ICP−MS. The method for Cd detection using ICP−MS was established according to previously reported methods with minor modifications (Egger et al., 2019; Tai et al., 2022). Tissues (100 mg) were homogenized in RIPA lysis buffer (1,000 μL) (P0013C; Beyotime Biotechnology, Shanghai, China) with a tissue grinder. Tissue homogenate was centrifuged at 4 °C, 5,000 rpm for 5 min. Then, 100 μL of supernatant or serum was digested with 68% HNO3 in 3.9 mL and reacted in a 15 mL centrifuge tube for 5 min. Subsequently, the reaction solutions were diluted with deionized water and analyzed using a 7800 ICP−MS instrument (Agilent Instruments, Tokyo, Japan). The optimum instrument conditions were set as listed in Table 1. The protein concentration of the supernatant was determined using a BCA kit (P0010S; Beyotime, Shanghai, China). The Cd concentration in each organ or tissue was expressed in unit of μg/mg protein, while the serum Cd concentration was expressed in units of μg/L and μg/mg protein.

Table 1 Operating parameters for the 7800 ICP-MS.

Parameters	Setting	
ICP radio-frequency power	1,550 W	
Plasma gas	15 L/min	
Auxiliary gas	0.9 L/min	
Nebulizer	1.07 L/min	
Helium flow	4.4 ml/min	

DAPI staining

Azoramide and Cd concentrations were selected according to previously reported works (Fu et al., 2015; Komoike, Inamura & Matsuoka, 2012; Zhang et al., 2019). HK-2 cells and ARPE-19 cells were incubated with azoramide (20 μM) for 5 h and then treated in the presence or absence of Cd (20 μM) for 24 h. After washing with phosphate-buffered saline (PBS) three times, the cells were fixed with 4% paraformaldehyde for 15 min at room temperature and stained with DAPI for 5 min in the dark. Cell nuclei were observed and photographed by fluorescence microscopy (IX53, Olympus, Tokyo, Japan). The healthy cell nucleus is uniformly stained and clear-edged, while the apoptotic nuclei show irregular edges around the nucleus, heavier staining, and nuclear pyknosis. According to the previously reported method (Jeon et al., 2023), the ratio of condensed nuclei to total nuclei was calculated and expressed as the apoptosis rate (% apoptosis) in each group, and at least three photos were included in the analysis.

Cell apoptosis analysis

Cell apoptosis was examined with flow cytometry (DxFLEX; Beckman Coulter, Suzhou, Jiangsu, China) following annexin V/propidium iodide (PI) double staining (CA1020; Solarbio, Beijing, China). After the indicated treatment, the cells were trypsinized and rinsed with PBS and then resuspended in 100 μL of binding buffer. Five microliters of annexin V and 5 μL of PI were sequentially added and incubated for 5 min in the dark at room temperature. Subsequently, cells were diluted with 400 μL binding buffer, and at least 104 cells were analyzed in each treatment. Annexin V-positive cells were regarded as apoptotic cells. Therefore, cells in quadrants 1 and 4 of the flow cytometry dot plots with quadrant markers were considered apoptotic. Apoptosis rate = (annexin V positive cell number/total cell number) × 100%.

Western blotting (WB)

Total proteins were extracted with RIPA lysis buffer (Beyotime, Shanghai, China), which contains 1 mM phenylmethanesulfonyl fluoride (ST506; Beyotime, Shanghai, China) and 1% protein phosphatase inhibitor cocktail (P1260; Solarbio, Beijing, China). The protein concentrations were determined with a BCA protein quantitation kit (P0010S; Beyotime, Shanghai, China). Next, 10 μg of protein was subjected to sodium dodecyl sulfate–polyacrylamide gel electrophoresis and transferred onto polyvinylidene difluoride membranes (ISEQ00010; Millipore, Burlington, MA, USA). After blocking with 5% nonfat milk at room temperature for 1 h, the membranes were incubated overnight at 4 °C with the indicated primary antibodies. Following this, the membranes were incubated with their corresponding horseradish peroxidase (HRP)-conjugated secondary antibodies (115-035-044 and 111-035-003, respectively; dilution: 1:2,500 and 1:5,000, respectively; Jackson ImmunoResearch Laboratories, West Grove, PA, USA) for 1 h at room temperature. Finally, the protein bands were visualized using the ChemiDox™ XRS+ system (Bio-Rad, Hercules, CA, USA) after exposure to Western ECL Substrate (1705061; Bio-Rad, Hercules, CA, USA) for 2 min. The optical density of each WB band was measured using Image Lab™ software (Bio-Rad, Hercules, CA, USA) and normalized to the densities of GAPDH for the same samples. The results are expressed as fold changes relative to the corresponding control, which was always considered as 1.

Mitochondria-associated reactive oxygen species (ROS) measurements

After the designated treatment, mitochondrial ROS were assessed by MitoSOX™ Red (M36008; Invitrogen, Thermo Fisher Scientific, Waltham, MA, USA) staining according to the manufacturer’s instructions. Cells were collected and resuspended in 100 μL DMEM/F12 medium and incubated with 2.5 μM MitoSOX™ Red in the dark at 37 °C for 15 min. After washed with PBS three times, the cells were analyzed by flow cytometry (DxFLEX), and at least 104 cells were included in the analysis. CytExpert software (Beckman Coulter, Suzhou, Jiangsu, China) was used for measurement of the mean fluorescence intensity of each sample, and the results were expressed as fold changes relative to the control group.

Quantitative real-time PCR (qPCR)

Total RNA was extracted and purified with a RNAiso Plus kit (9109; TaKaRa bio-INC., Kusatsu, Shiga, Japan). cDNA was synthesized with a PrimeScript™ RT Master Mix Kit (RR036A; TaKaRa, Kusatsu, Shiga, Japan). Then, qPCR detection was conducted on an ABI Prism® 7500 real-time PCR detection system (Applied Biosystems, Thermo Fisher Scientific, Waltham, MA, USA) with a TB Green® Premix Ex Taq™ (Tli RNaseH Plus) kit (RR420A; TaKaRa, Kusatsu, Shiga, Japan). The transcriptional expressions of C/EBP homologous protein (CHOP) and heme oxygenase 1 (HO-1) were measured by qPCR. The primers used in the present study were as follows: HO-1, 5′-CCAGCGGGCCAGCAACAAAGTGC-3′, 5′-AAGCCTTCAGTGCCCACGGTAAGG-3′; CHOP, 5′-GACCTGCAAGAGGTCCTGTC-3′, 5′-TGTGACCTCTGCTGGTTCTG-3′; and GAPDH, 5′-TGACGCTGGGGCTGGCATTG-3′ and 5′-GGCTGGTGGTCCAGGGGTCT-3′. Relative gene expression was calculated with the 2−ΔΔCt method with GAPDH as a loading control. At least three independent samples were performed in each group, and each sample was measured in triplicate.

JC-1 staining

Disruption of mitochondrial inner transmembrane potential ( ΔΨ m) triggers the activation of intrinsic apoptotic pathway. In this study, ΔΨ m was evaluated using JC-1, a dye that accumulates in mitochondria in a potential-dependent manner. When mitochondrial membrane integrity is compromised, the fluorescence emission of JC-1 shifts from red (aggregate) to green (monomer) (Li et al., 2016). The JC-1 staining kit (C2006) from Beyotime Company was used for this purpose. HK-2 cells and ARPE-19 cells were incubated with azoramide (20 μM) for 5 h and then treated with or without Cd (20 μM) for 24 h. JC-1 staining was performed according to the manufacturer’s instructions. Briefly, cells were washed with PBS, incubated with JC-1 for 30 min at 37 °C, and observed under an inverted fluorescence microscope (IX53, Olympus, Tokyo, Japan).

Inhibition of p38 MAPK and JNK

HK-2 cells and ARPE-19 cells were seeded in 6-well plates and incubated with Cd (20 μM) in the presence or absence of SB203580 (10 μM) or SP600125 (10 μM) for 24 h. Concentrations of the inhibitors were selected based on previous studies (Jin et al., 2023; Tang et al., 2023). Then, the cells were collected for WB analysis or stained with annexin V/PI and measured by flow cytometry.

Statistical analysis

Data are expressed as the means ± standard deviations (SD). For statistical analysis, the unpaired Student’s t test was used for data with only two groups; one-way analysis of variance followed by Tukey’s multiple comparisons test was used for data containing more than two groups. All data were analyzed with GraphPad Prism software (version 9.0; San Diego, CA, USA), and p < 0.05 was considered statistically significant.

Results

Accumulation of Cd in different tissues of ICR mice

We observed significant body weight loss in ICR mice after repeated exposure to Cd for 7 days (Fig. 1A and Table 2). However, Cd exposure did not result in any statistically significant differences in liver (1.24 ± 0.14 g vs 1.14 ± 0.09 g), heart (0.12 ± 0.01 g vs 0.12 ± 0.04 g), or lung (0.15 ± 0.02 g vs 0.16 ± 0.02 g) weights (Table 2). Nevertheless, Cd exposure led to increased spleen weight (0.06 ± 0.01 g vs 0.15 ± 0.01 g) while inducing weight loss in the kidney (0.39 ± 0.02 g vs 0.34 ± 0.02 g) and testis (0.16 ± 0.01 vs 0.07 ± 0.04 g) (Table 2). The heart coefficient, liver coefficient, and kidney coefficient were not significantly altered (Figs. 1B–1D). However, the lung coefficient and spleen coefficient showed significant increases (Figs. 1E and 1F). Notably, the testis coefficient showed a dramatic decrease following Cd treatment (Fig. 1G).

Figure 1 Accumulation of Cd in different tissues of ICR mice.

ICR mice were intraperitoneally administered CdCl2 (5 mg/kg) once a day for 7 days. (A) Mouse body weights were recorded before (on Day 0) and after (on Day 7) Cd treatment (n = 5). (B–G) The ratios (%) of organ weights (heart, liver, kidney, lung, spleen, and testis) to body weights were calculated as organ coefficients and measured after Cd treatment for 7 days (n = 4 or 5). (H and I) Cd concentration in the serum was measured with ICP‒MS and expressed as μg/L or μg/g protein (n = 4 or 5). (J) Cd concentration in the eyes was measured with ICP‒MS (n = 5). (K) The Cd level in various tissue samples was detected by ICP‒MS (n = 4 or 5). *p < 0.05, **p < 0.01, ***p < 0.001 for the indicated comparisons.

Table 2 Body and organ weights of the mice.

	Control	Cadmium	p	
Initial body weight (g)	22.20 ± 1.35	24.12 ± 1.53	0.0686	
Last body weight (g)	22.50 ± 1.43	20.68 ± 0.78*	0.0370	
Liver weight (g)	1.24 ± 0.14	1.14 ± 0.09	0.2250	
Heart weight (g)	0.12 ± 0.01	0.12 ± 0.04	0.9100	
Kidney weight (g)	0.39 ± 0.02	0.34 ± 0.02**	0.0083	
Lung weight (g)	0.15 ± 0.02	0.16 ± 0.02	0.6351	
Spleen weight (g)	0.06 ± 0.01	0.15 ± 0.01***	<0.0001	
Testis weight (g)	0.16 ± 0.01	0.07 ± 0.04**	0.0011	
Notes:

Values are expressed as mean ± SD.

* p < 0.05.

** p < 0.01.

*** p < 0.001 compared with the control group.

To assess Cd concentration in various organs of ICR mice, we employed ICP-MS. Our findings demonstrated that Cd concentration in the serum of control mice was 2.95 ± 5.46 μg/L (or 0.04 ± 0.08 μg/g protein) (Figs. 1H and 1I). In the Cd exposure group, Cd concentration significantly increased to 44.22 ± 11.27 μg/L (or 0.72 ± 0.17 μg/g protein) in comparison to the control (Figs. 1H and 1I). Notably, we found that Cd accumulated in the eye at a higher concentration (10.83 ± 9.14 μg/g protein vs 2.11 ± 0.62 μg/g protein in the control) (Fig. 1J). Among the organs tested, Cd concentration ranked highest in the liver (741.68 ± 81.65 μg/g protein), followed by the kidney (358.35 ± 28.24 μg/g protein), testis (153.52 ± 45.67 μg/g protein), lung (45.69 ± 19.65 μg/g protein), spleen (37.61 ± 13.11 μg/g protein), eye (10.83 ± 9.14 μg/g protein), and serum (0.72 ± 0.17 μg/g protein) (Fig. 1K). These data indicate that Cd tends to accumulate in multiple vital organs, with a preference for the liver and kidney. Intriguingly, the testis showed higher Cd concentration compared to the lung or spleen, confirming that Cd is reproductively toxic (Fig. 1K).

Azoramide protects cells against Cd-induced toxicity

To investigate the protective role of azoramide in Cd-induced cytotoxicity, two widely used cell types, the human proximal tubule epithelial cell line HK-2 and the human retinal pigment epithelial cell line ARPE-19, were employed in the study. We found that Cd exposure could significantly induce apoptotic nuclear morphological changes, and the proportion of nuclear shrinkage and the intensity of DAPI staining increased significantly (Figs. 2A and 2B). Cleavage of PARP1 (C-PARP1), a widely used apoptotic marker protein, was elevated after Cd treatment in HK-2 cells (Fig. 2C). In contrast, azoramide coincubation significantly suppressed PARP1 cleavage (Fig. 2C), indicating that azoramide exerts a protective effect on Cd-exposed HK-2 cells. However, the expression of C-PARP1 was too low to be detectable in ARPE-19 cells in the present study. We further detected apoptosis by annexin V/PI double-staining. Cd treatment dramatically increased the cell apoptosis rate compared to the control group (Figs. 2D and 2E). However, azoramide significantly alleviated Cd-induced cell apoptosis in HK-2 and ARPE-19 cells, and the apoptosis rate was reduced by nearly 20% (Figs. 2D and 2E). Overall, these results suggest that azoramide can significantly protect against Cd-induced cytotoxicity.

Figure 2 Azoramide protects cells against Cd-induced toxicity.

(A and B) HK-2 and ARPE-19 cells were stained with DAPI, and representative images were captured under an inverted fluorescence microscope. The ratio of apoptotic nuclei to total nuclei was calculated and expressed as the apoptosis rate; each value is presented as the mean ± SD of at least three fields of view under the microscope. (C) HK-2 cells were pretreated with 20 μM azoramide for 5 h and coincubated with Cd (20 μM) for 24 h. PARP1 and C-PARP1 were detected by WB, and corresponding densitometric analysis was conducted (n = 3). (D and E) HK-2 and ARPE-19 cells were preincubated with azoramide (20 μM) for 5 h and subsequently treated with or without Cd (20 μM) for 24 h. Cell apoptosis was assessed with flow cytometry after annexin V/PI staining (n = 3 or 6). **p < 0.01, ***p < 0.001 for the indicated comparisons.

Azoramide inhibits Cd-induced ER stress

Azoramide has been shown in previous studies to be a small molecule inhibitor of ER stress (Fu et al., 2015; Ke et al., 2020). In our investigation, we found that azoramide treatment significantly enhanced Cd-induced BiP expression (Figs. 3A and 3B). Additionally, our results showed that Cd increased the phosphorylation level of PERK, whereas azoramide incubation effectively attenuated this increase (Figs. 3C and 3D). Our data showed that Cd increased p-eIF2α expression in HK-2 and ARPE-19 cells (Figs. 3E and 3F). However, azoramide exacerbated the phosphorylation of eIF2α (Figs. 3E and 3F), which may inhibit the translation process and alleviate the pressure of protein folding and processing within the ER. Moreover, our study revealed that Cd treatment elevated the mRNA expression of CHOP, a key mediator of programmed cell death (Figs. 3G and 3H). Azoramide significantly inhibited the upregulation of CHOP (Figs. 3G and 3H). These results strongly suggest that the azoramide’s protective effect against Cd toxicity is closely associated with its ability to inhibit ER stress.

Figure 3 Azoramide inhibits Cd-induced ER stress.

(A–F) HK-2 and ARPE-19 cells were preincubated with azoramide (20 μM) for 5 h and subsequently treated with or without Cd (20 μM) for 24 h. BiP, p-PERK, and p-eIF2α were detected by WB, and the corresponding densitometric analysis was performed (n = 3, 4, or 5). (G and H) HK-2 and ARPE-19 cells were preincubated with azoramide (20 μM) for 5 h and then treated with or without Cd (15 μM) for 24 h. CHOP expression was detected by qPCR (n = 3). *p < 0.05, **p < 0.01, ***p < 0.001 for the indicated comparisons.

Azoramide inhibits Cd-induced intracellular oxidative stress

It is well-known that ROS can be generated from oxidative phosphorylation in mitochondria (Li et al., 2015). As byproducts in mitochondria, excess ROS may disrupt the electron transport chain and result in mitochondrial dysfunction. We therefore measured mitochondrial ROS with MitoSOX red staining and found that Cd exposure significantly upregulated ROS production in mitochondria, while azoramide suppressed mitochondrial ROS elevation (Figs. 4A and 4B). The Nrf2 pathway is one of the major cellular defense mechanisms against oxidative stress (Tonelli, Chio & Tuveson, 2018). To elucidate its role in azoramide-induced protection effects, we detected Nrf2 by WB. We found that Cd exposure promoted Nrf2 expression, whereas azoramide treatment diminished Cd-induced Nrf2 upregulation (Figs. 4C and 4D). As a consequence, the expression of downstream targeted genes, such as HO-1, was inhibited in the presence of azoramide (Figs. 4E and 4F). Furthermore, JC-1 staining revealed that normal cells exhibited an intact ΔΨ m, whereas Cd treatment induced markedly ΔΨ m loss, as indicated by a shift in JC-1 fluorescence from red to green (Figs. 4G and 4H). Azoramide, however, dramatically attenuated Cd-induced ΔΨ m loss and promoted cell survival (Figs. 4G and 4H). Collectively, these data suggest that azoramide may suppress ROS production in mitochondria and inhibit intracellular oxidative stress, thus inhibiting Cd-induced mitochondrial injury.

Figure 4 Azoramide inhibits Cd-induced intracellular oxidative stress.

HK-2 and ARPE-19 cells were preincubated with azoramide (20 μM) for 5 h and subsequently treated with or without Cd (20 μM) for 24 h. The cells were then stained with MitoSOX and analyzed by flow cytometry (n = 3 or 5) (A and B); the expression of Nrf2 was measured by WB (n = 3) (C and D). HK-2 and ARPE-19 cells were preincubated with azoramide (20 μM) for 5 h and then treated with or without Cd (15 μM) for 24 h. The expression of HO-1 was detected by qPCR (n = 3 or 6) (E and F). (G and H) After treatment with Cd (20 μM) and azoramide (20 μM), mitochondrial function in HK-2 and ARPE-19 cells was determined by staining with the ΔΨm sensitive dye JC-1. Representative images were captured under an inverted fluoresce microscope. **p < 0.01, ***p < 0.001 for the indicated comparisons.

Azoramide inhibits the activation of p38 MAPK and JNK signaling

Evidence suggests that the MAPK signaling pathway may play a role in Cd-induced cell damage (Kalariya et al., 2009). In our study, we observed that Cd treatment significantly increased the phosphorylation levels of p38 MAPK and JNK (Figs. 5A–5D). However, azoramide dramatically inhibited p-p38 MAPK and p-JNK expression, suggesting that azoramide may inhibit the activation of MAPK signaling (Figs. 5A–5D). These findings also indicate that ER stress, as inhibited by azoramide, may be an upstream event in Cd-induced MAPK activation. To clarify the role of p38 MAPK and JNK in Cd-induced cell death in HK-2 and ARPE-19 cells, cells were treated with pharmacological inhibitors. We found that Cd-induced PARP1 cleavage in HK-2 cells was diminished by SB203580, an inhibitor of p38 MAPK (Fig. 5E). Conversely, the JNK inhibitor SP600125 significantly promoted Cd-induced cleaved PARP1 expression (Fig. 5F). Consistently, apoptosis analysis showed that SB203580 decreased Cd-induced apoptosis rate in HK-2 and ARPE-19 cells (Figs. 5G and 5I), whereas SP600125 aggravated Cd-induced cell apoptosis (Figs. 5H and 5J). These results suggest that p38 MAPK activation is toxic to HK-2 and ARPE-19 cells. In contrast, JNK activation is beneficial for cell survival.

Figure 5 Azoramide inhibits the activation of p38 MAPK and JNK signaling.

(A and B) HK-2 cells were preincubated with azoramide (20 μM) for 5 h and subsequently treated with or without Cd (20 μM) for 24 h. The expression levels of p-38 and p-JNK were detected by WB, and corresponding densitometric analysis was conducted (n = 3). (C and D) ARPE-19 cells were preincubated with azoramide (20 μM) for 5 h and subsequently treated with or without Cd (20 μM) for 24 h. Then, p-38 and p-JNK were detected by WB, and corresponding densitometric analysis was performed (n = 3 or 4). (E and F) HK-2 cells were incubated with Cd (20 μM) in the presence or absence of the p38 MAPK inhibitor SB203580 (10 μM) or the JNK inhibitor SP600125 (10 μM) for 24 h. C-PARP1 was measured by WB, and corresponding densitometric analysis was conducted (n = 3). (G–J) HK-2 and ARPE-19 cells were incubated with Cd (20 μM) in the presence or absence of the p38 MAPK inhibitor SB203580 (10 μM) or the JNK inhibitor SP600125 (10 μM) for 24 h, and then cell apoptosis rates were analyzed with flow cytometry (n = 3). *p < 0.05, **p < 0.01, ***p < 0.001 for the indicated comparisons.

Azoramide inhibits Cd-induced autophagy

Autophagy may also play a certain role in Cd-induced cell apoptosis (Zhang et al., 2019). Our study found that Cd exposure significantly upregulated the expression of autophagy-related marker proteins, such as BECN1 and LC3BII (Fig. 6). However, azoramide treatment inhibited Cd-induced BECN1 expression, and markedly suppressed LC3BII elevation in HK-2 and ARPE-19 cells (Fig. 6). These findings suggest that azoramide alleviates Cd-induced autophagy.

Figure 6 Azoramide inhibits Cd-induced autophagy.

HK-2 and ARPE-19 cells were preincubated with azoramide (20 μM) for 5 h and subsequently treated with or without Cd (20 μM) for 24 h. (A and D) The expression levels of BECN1 and LC3B were determined by WB. (B, C, E, and F) The corresponding densitometric analysis was performed (n = 3 or 4). *p < 0.05, **p < 0.01, ***p < 0.001 for the indicated comparisons.

Discussion

Humans are exposed to Cd by consuming contaminated food and drink. As a nonbiodegradable heavy metal, Cd exposure may lead to fatal implications for life (Genchi et al., 2020). In this study, we measured Cd concentrations in different organs by using ICP‒MS. Our findings demonstrate that the liver and kidney may be two preferable organs for Cd accumulation in the body, which is consistent with previous studies conducted in mice or humans (Egger et al., 2019; Tai et al., 2022; Winiarska-Mieczan & Kwiecien, 2016). Additionally, we observed that the Cd concentration in the testis was higher than in the lung and spleen. Accumulated evidence suggests high sensitivity of the testes to Cd (Ali et al., 2022). Furthermore, our study suggests that the relatively high accumulation of Cd in the testis may also contribute to testicular atrophy (weight loss). We also confirmed that Cd accumulation in the eye, supporting the notion that Cd may be involved in the development of certain ocular diseases. Prior studies have shown that Cd accumulation in the retina may be closely related to the pathogenesis of age-related macular degeneration (Kalariya et al., 2009; Zhang et al., 2019).

Accumulation of Cd in various organs leads to tissue injury and organ dysfunction. Treatment strategies for Cd poisoning include promoting Cd excretion with chelating agents and attenuating its toxic effects using small molecules like melatonin and morin (Annie et al., 2023; Xie et al., 2022). Cd has been shown to disrupt cell metabolic activities and induce excessive accumulation of ROS, resulting in cell apoptosis (Zhang et al., 2019, 2020; Zhao et al., 2021). Emerging evidence indicates that the ER may be the target organelle for Cd toxicity (Zhang et al., 2019; Zhao et al., 2021), and inhibition of ER stress has been identified as an effective strategy against Cd-induced cytotoxicity. Previous studies have reported that certain ER stress inhibitors, such as tauroursodeoxycholic acid and salubrinal, exert protective effects against Cd-induced cell injury (Chen et al., 2019; Komoike, Inamura & Matsuoka, 2012). However, these small molecules may lack mechanistic specificity and have limited success in clinical applications (Hetz, Chevet & Harding, 2013). By contrast, azoramide, an ER-targeted small-molecule compound, has been found to activate ER chaperone capacity and protect cells against ER stress (Fu et al., 2015). Our study demonstrates that azoramide has the potential to serve as an effective therapeutic agent against Cd-induced toxicity.

Azoramide significantly mitigated Cd-induced cell apoptosis in both HK-2 and ARPE-19 cells, suggesting that its protective effect is independent of cell type. Our results also indicate that azoramide treatment attenuated Cd-induced Nrf2 activation, suggesting that it may not exert its protective mechanism through Nrf2 signaling. Considering that Nrf2 expression is regulated by intracellular ROS levels (Tonelli, Chio & Tuveson, 2018), our findings suggest that azoramide may suppress Nrf2 activation by inhibiting ROS generation.

When unfolded proteins accumulate in the lumen during ER stress, PERK dissociates from the ER-resident molecular chaperone BiP and is autophosphorylated (Li et al., 2015). PERK phosphorylates eIF2α and hampers the global protein synthesis, thus relieving ER stress. In addition, p-eIF2α selectively enhances the translation of activating transcription Factor 4 (ATF4), which activates the expression of genes involved in antioxidant responses, amino acid biosynthesis, and transport to reestablish cell hemostasis and promote cell survival (Zhang et al., 2022). Our data showed that azoramide enhanced Cd-induced BiP expression, which boosts ER protein folding acutely and binds with PERK and further inhibits Cd-induced PERK phosphorylation. It is important to note that azoramide does not fully increase chaperone capacity without PERK activity, despite attenuating stress-induced PERK expression (Fu et al., 2015; Miao et al., 2022; Okatan et al., 2019). We observed that azoramide treatment dramatically promotes Cd-induced eIF2α phosphorylation and induces cell protective effects, which aligns with previously reported studies (Fu et al., 2015). It should be emphasized that multiple eIF2α kinases, including PERK, GCN2 (general control non-derepressible-2), PKR (double-stranded RNA activated protein kinase), and HRI (heme-regulated inhibitor), can affect the phosphorylation of eIF2α in response to different stresses (Halliday, Hughes & Mallucci, 2017). Therefore, the azoramide-associated eIF2α phosphorylation observed in our study may be induced by the activation of other eIF2α kinases, excluding PERK.

Acute and severe ER stress can induce cell apoptosis through the upregulation of CHOP, resulting in the downregulation of the antiapoptotic protein B-cell lymphoma-2 (Bcl-2) and increased production of the proapoptotic protein Bcl-2 interacting mediator of cell death (Bim) (Fu et al., 2015; Li et al., 2015). A study has confirmed that azoramide can protect cell from CHOP induction (Fu et al., 2015). Consistent with these findings, our study demonstrates that azoramide dramatically mitigates ER stress and significantly suppresses Cd-induced CHOP expression. Additionally, we observed that azoramide, as an ER stress inhibitor, mitigated Cd-induced oxidative stress. This suggests that Cd-activated ER stress may trigger ROS generation in mitochondria. Moreover, CHOP can upregulate endoplasmic oxidoreductin-1 (ERO1), further enhancing intracellular oxidative stress (Ong & Logue, 2023). ERO1 catalyzes the reduction of oxygen into hydrogen peroxide (H2O2) during oxidative protein folding. The increase in ERO1 and H2O2 contributes to the upregulation of ROS-sensitive Ca2+ channels, such as the inositol-1,4,5-trisphosphate receptor (IP3R), resulting in increased Ca2+ leakage from the ER to mitochondria. This process promotes metabolism and aggravates oxidative stress (Cao & Kaufman, 2014; Decuypere et al., 2011; Marciniak et al., 2004). Furthermore, the diffusion of ROS from mitochondria can attack the ER, further exacerbating ER stress (Decuypere et al., 2011). Thus, a vicious cycle between the ER and mitochondria is formed, triggering cell apoptosis. Consequently, our study demonstrates that azoramide treatment alleviates oxidative stress by restoring ER hemostasis, ultimately inhibiting mitochondrial dysfunction and cell apoptosis.

JNK and p38 MAPK are preferentially activated in response to environmental stress, particularly oxidative stress (Cao et al., 2014). Previous studies have suggested that Cd induces activation of JNK and p38 MAPK, but their roles in cell survival remain unclear (Kalariya et al., 2009). This study clarifies that p38 MAPK signaling promotes Cd-induced cell apoptosis, while the JNK signaling pathway attenuates Cd-induced cytotoxicity. These findings indicate that inhibiting p38 MAPK and activating JNK may be a promising approach to counter Cd-induced cytotoxicity. Furthermore, our study demonstrates that azoramide significantly reduces JNK and p38 MAPK phosphorylation. Since oxidative stress is an upstream event that triggers JNK and p38 MAPK signaling, and azoramide can decrease ROS generation, we conclude that the inhibitory effects of azoramide on JNK and p38 MAPK may be attributed to its suppression of ER stress-associated oxidative stress. Additionally, IRE1α is known to participate in the regulation of JNK and p38 MAPK signaling pathways (Zhang et al., 2022). However, it is worth noting that IRE1α is also closely involved in azoramide action (Fu et al., 2015), suggesting that IRE1α signaling may not specifically contribute to azoramide-induced effects on JNK and p38 MAPK signaling.

Autophagy is a catabolic mechanism that regulates the recycling and turnover of intracellular constituents in response to cell stress. ER stress has been identified as an upstream event in the activation of autophagy (Zhang et al., 2019, 2020). The three branches of the UPR modulate the initiation of autophagy through distinct pathways. For instance, CHOP can inhibit the mammalian target of rapamycin complex 1 (mTORC1) by activating AMP-activated protein kinase (AMPK) or tribbles homolog 3 (TRB3), thereby promoting unc-51-like kinase 1 (ULK1) complex formation and initiating autophagy (Zhang et al., 2022). Our previous data have also confirmed that autophagy is a mechanism of Cd-induced cytotoxicity (Zhang et al., 2019). Consistent with these findings, the present study reveals that the inhibition of ER stress using azoramide suppresses autophagy, potentially contributing to the protection against Cd-induced toxicity.

Several major limitations exist in this study. One limitation is that while we observed that p38 MAPK and JNK signaling may exert detrimental and beneficial roles, respectively, in cell survival upon Cd exposure, the underlying molecular mechanisms remain unclear. Another weakness is that the protective effect of azoramide on the kidneys and retinas in the context of Cd accumulation in vivo still requires further verification.

Conclusions

In summary, the current study suggests that azoramide has the potential to be a protective agent for treating Cd poisoning. Our findings provide insights into the molecular mechanisms underlying the protective effects of azoramide, including the suppression of ER stress, reduction of ROS generation, and attenuation of the MAPK cascade and autophagy activation. Based on this evidence, further exploration of azoramide as a therapeutic drug for Cd poisoning is warranted.

Supplemental Information

Supplemental Information 1 Raw data.

Files in .PZFX format can be accessed using GraphPad Prism software (version 9.0; San Diego, CA, USA) which can be downloaded from https://www.graphpad.com/.

Click here for additional data file.

Supplemental Information 2 ARRIVE 2.0 checklist.

Click here for additional data file.

Additional Information and Declarations

Competing Interests

Author Contributions

Ethics

Field Study Permissions

Data Availability

The authors declare that they have no competing interests.

Lingmin Zhang performed the experiments, analyzed the data, prepared figures and/or tables, authored or reviewed drafts of the article, and approved the final draft.

Jianguo Zhang performed the experiments, analyzed the data, prepared figures and/or tables, authored or reviewed drafts of the article, and approved the final draft.

Yingying Zhou performed the experiments, analyzed the data, prepared figures and/or tables, authored or reviewed drafts of the article, and approved the final draft.

Qingqing Xia performed the experiments, authored or reviewed drafts of the article, and approved the final draft.

Jing Xie performed the experiments, authored or reviewed drafts of the article, and approved the final draft.

Bihong Zhu analyzed the data, prepared figures and/or tables, authored or reviewed drafts of the article, and approved the final draft.

Yang Wang analyzed the data, prepared figures and/or tables, authored or reviewed drafts of the article, and approved the final draft.

Zaixing Yang conceived and designed the experiments, authored or reviewed drafts of the article, and approved the final draft.

Jie Li conceived and designed the experiments, performed the experiments, analyzed the data, prepared figures and/or tables, authored or reviewed drafts of the article, and approved the final draft.

The following information was supplied relating to ethical approvals (i.e., approving body and any reference numbers):

The Animal Care and Use Committee of Hangzhou Hibio technology Company.

The following information was supplied relating to field study approvals (i.e., approving body and any reference numbers):

All animal protocols were approved by the Animal Care and Use Committee of Hangzhou Hibio technology Company.

The following information was supplied regarding data availability:

The raw measurements are available in the Supplemental Files.

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
