# Peer review of "Azoramide ameliorates cadmium-induced cytotoxicity by inhibiting endoplasmic reticulum stress and suppressing oxidative stress"

_PeerJ, doi:10.7717/peerj.16844_

## Round 0.1 · original submission · Major Revisions

The manuscript #88215 offers a compelling exploration into the potential of azoramide to alleviate cadmium-induced cytotoxicity by targeting endoplasmic reticulum (ER) stress and oxidative stress pathways. While the study's findings are promising, there are several critical points that require careful consideration and revision.

An important concern arises from the lack of comprehensive methodology details, which hinders reproducibility. The manuscript should provide in-depth protocols for key experiments, including JC-1 staining and inhibitor treatments, to ensure accurate replication. Furthermore, the rationale behind the chosen concentrations of cadmium and azoramide needs explicit clarification, along with maintaining consistency between reported and utilized concentrations.

Data presentation also requires refinement. To enhance transparency, quantified data should accompany the presented Western blot images. Additionally, uniform protein nomenclature, especially for eIF2α, should be maintained. These revisions would provide a clearer representation of the experimental outcomes.

The study's conclusions should align more precisely with the data presented. Conflicting interpretations of results, such as the relationship between azoramide, ER stress, and autophagy, should be clarified to enhance the coherence of the manuscript. Addressing these concerns would strengthen the manuscript's scientific rigor and its contribution to the field.

Overall, a major revision based on the comments of both of the reviewers is recommended. By addressing these issues, the manuscript's impact and significance would be better realized, and its potential as an insightful study into azoramide's therapeutic potential against cadmium-induced cytotoxicity would be duly affirmed.

**Language Note:** The review process has identified that the English language must be improved. PeerJ can provide language editing services - please contact us at copyediting@peerj.com for pricing (be sure to provide your manuscript number and title). Alternatively, you should make your own arrangements to improve the language quality and provide details in your response letter. – PeerJ Staff

Reviewer 1 ·

Basic reporting

English language needs to be improved, some literature reference is missing

Experimental design

Experimental design is good with appropriate controls and statistics.

Validity of the findings

Conclusions are mostly well defined, however in some cases over statements have been made which are not supported by the present data.

Additional comments

Can be improved by addressing the comments

Annotated reviews are not available for download in order to protect the identity of reviewers who chose to remain anonymous.

·

Basic reporting

• Line 124, delete full stop after pyknosis
• Line 198. We found that azoramide can significantly enhanced Cd-induced Bip expression- rewrite the sentence.
• Line 245- Previously reported study has demonstrated that Cd accumulation in retina may be closely implicated in the pathogenesis of age-related macular degeneration. Please add reference.
• Line 258- We found azoramide was potential to be an effective therapeutic agent. Please rewrite the sentence.
• Line 210- It is well-known that ROS can generated from oxidative phosphorylation in mitochondria. Please rewrite the sentence.
• Line 214- please correct to figure 4 A and E
• For figure 5C, please follow the same order of 5D for different treatment groups.
• Authors have used eIF2α in some places and eIF2a in other places. Please make it uniform.
• Please expand Bim and Bip as they first appear in the manuscript.
• Line 239- Our study demonstrated that liver and kidney may respectively be the first and second preferable organ in the body, these findings were in consistence with previously reported researches. Please rewrite the sentence.
• Line 271. “Azoramide significantly inhibited Cd-induced CHOP expression, meanwhile, we found that azoramide inhibited Cd-induced oxidative stress, suggesting that Cd-activated ER stress initiates ROS generation in mitochondria”- please split into 2 sentences.
• The manuscript does not have page numbers.
• Please check the references for uniformity. Some references are in sentence case and others are not.

Experimental design

• Methods section is missing many details. There is no reference given for the methodologies used in the study. Authors should explain JC-1 staining protocol in the methods section. Authors should give details such as concentration, duration etc. regarding the treatment of cells with p38 or JNK inhibitors. Authors should include the amount of protein used for western blot for different proteins.
• How did the authors determine the dosage of Cadmium for the in vivo study? Please provide detailed information in the methods section. Additionally, what criteria were employed in selecting the concentrations of Cd and azoramide for the in vitro study? According to the methods section (line 120), the authors utilized 20 µM azoramide and 15 µM Cd for the study. However, the results presented indicate the use of 20 and 30 µM azoramide, and 10, 15, and 20 µM Cd. The authors should elucidate the rationale behind employing different concentrations of Cd and azoramide in the study.

Validity of the findings

• Authors have given graphs with individual data points in figure 1 A-G. Similarly, authors should include graphs with individual data points for all the graphs in the manuscript. In certain experiments, the authors utilized n=3 per group; however, it is recommended that authors increase this to n=5-6 per group.
• A major part of the study is determination of different signaling pathways using western blot. However, the Western blot results are presented solely as images, without corresponding quantification data. Loading control GAPDH appears inconsistent across different samples. For e.g. figure 3A, Cd shows increased Bip. However, the GAPDH band in control is much weaker compared to the Cd group GAPDH band. Similarly, check figure 4F and 6B. In order to see the actual differences between treatment groups, authors must include western blot quantification data.
• Authors detect Cd in control mice blood at 2.95±5.46 µg/L. Experimental animals are kept in controlled environments to minimize external factors that could influence the results of experiments. Does the Cd level in control mice mean that these mice are exposed to contaminated food, air or water?
• Figure. 2 F- Was the total cells sampled same in all groups? Which quadrants are being considered apoptotic - label quadrants or write in methods how apoptosis rate is calculated. Apoptosis rate? Word usage suggests apoptosis over period of time; could also report as just % apoptosis.
• The gene expression data in figures 4C and 4G shows zero in control groups. Could you please provide an explanation for the absence of gene expression data in the control groups in figures 4C and 4G?

Additional comments

• Induction of HO-1 by NRF2 is a key mechanism through which cells counteract oxidative stress and other stressors. However, as per figure 4, azoramide diminished Nrf2 upregulation, and inhibited the expression of downstream target gene HO-1. How could this be considered as an indicator of azoramide induced protection? Please explain in the discussion section.
• Line 268, authors state that “Phosphorylation of eIF2a decreases global protein synthesis, but increase the pro-apoptotic protein CHOP expression (Fu et al. 2015). CHOP promotes cell apoptosis through decreasing the expression of Bcl-2 and increasing the Bim production”. This means that azoramide increases phosphorylation of eIF2α, which inhibit protein synthesis and protects from ER stress. At the same time, eIF2α increases CHOP expression, which can promote apoptosis. However, authors found that azoramide significantly inhibited Cd-induced CHOP expression (figure 3 G). How does azoramide increases eIF2a phosphorylation, but inhibits CHOP expression? This is opposite to what is written in line 268. Please re-write the discussion to make it more clear.

---

## Round 0.2 · Minor Revisions

The editor expresses satisfaction in observing significant improvements in PeerJ manuscript #88215, titled "Azoramide mitigates cadmium-induced cytotoxicity by inhibiting endoplasmic reticulum stress and suppressing oxidative stress", following the authors' diligent attention to the major concerns raised by both reviewers. Nonetheless, minor concerns identified by the reviewers persist, necessitating further refinement before the manuscript can be accepted for publication. The authors are kindly urged to thoroughly address these remaining comments and concerns from both reviewers to ensure the manuscript meets the necessary standards for acceptance.

Reviewer 1 ·

Basic reporting

The revised manuscript has improved considerably by addressing the concerns. However, there are still some concerns which should be taken care of before it is accepted.

1. In Fig. 1A the labelling of the panels is incorrect. The body weights shown in the figure (control group day 0 vs day 7 values shown almost same) does not correspond with Table 2 (values mentioned are 22.2 vs 24.12).

2. Can the authors explain how is weight loss in kidney after Cd treatment significant compared to non-treated. It is said in line 233 that there is no difference in weight of liver (a reduction of 0.1g) whereas in kidney there is a difference (reduction of 0.05g). Also, the kidney index change is not significant, so how weight change in kidney is significant?

Experimental design

Experiments have been designed and performed addressing the research question. Methodology has been described in detail.

Validity of the findings

Conclusions are valid. There is one minor concern. In Discussion (line 327), it is mentioned that the present study confirmed that Cd can accumulate in the eyes, however no data has been presented for this. The conclusion should accompany a data or should be clarified that no data has been presented.

Additional comments

4. Line 252: The result section starts with ‘to verify the role of azoramide in Cd induced cytoxicity’. This should be re-written to imply the protective role of azoramide rather than implying otherwise. Also, the study is not aimed at ‘verification’ of a previous reported protective effect of azoramide on Cd cytoxicity, so ‘investigate’ would be a more suitable choice of word.

·

Basic reporting

• Line 131. DAPI staining. Authors should cite reference where the ratio of condensed nuclei to total nuclei is used to calculate apoptosis.
• Line 223. Correct Fig.I to Fig.1I
• Authors are recommended to follow the same order for different groups in the figures. For e.g. figure 2A, the order of groups in the bar graphs are Con, Azo, Cd, Azo+Cd. In figure 2C, the order is Con, Cd, Azo, Azo+Cd. Similarly, Fig.3 A-F has one order and G-H has another order. Please make it uniform throughout the manuscript.
• In figure legends 3 and 4, authors write that Cd (15 or 20 µM). However, as per the labels above the figures, authors have used Cd only 20 µM. Please correct the legends for Cd concentration.

Experimental design

• Throughout the study, authors have repeated measurements in HK-2 and AR-19 cells. However, in figure 5E and F, the effect of inhibitors on C-PARP1 were studied on HK-2 cells while the apoptosis with the same inhibitors were determined on ARPE-19 cells. Authors are recommended to measure both C-PARP1 and apoptosis with the inhibitors in the same cell line.

Validity of the findings

No comment

Additional comments

• One of my previous suggestion was to include graphs with individual data points. However, authors have not included graphs with data points in the revised version.
• There requires more clarity regarding the following statement. Line 344. “Our study concluded that azoramide significantly attenuated JNK and p38 MAPK phosphorylation, which may be attributed to its inhibition of ER stress-associated oxidative stress”. As per figure 5G, inhibition of MAPK protected against cell death while inhibition of JNK increased cell death. Considering this, is it correct to state that azoramaide attenuated JNK phosphorylation and protected against oxidative stress?
• As per the discussion, PERK phosphorylates eIF2α which enhances ATF4 and antioxidant responses. Line 319, authors state that “azoramide enhanced Cd-induced BiP expression, which boosts ER protein folding acutely and binds with PERK and further inhibits Cd-induced PERK phosphorylation”. If azoramide inhibits PERK phosphorylation in Cd + azoramide group, how does it promote eIF2α phosphorylation in the same group. Please explain in the discussion section.

---

## Round 0.3 · accepted · Accept

The authors have satisfactorily addressed the comments of the reviewers and the manuscript is now of acceptable quality.

Reviewer 1 ·

Basic reporting

No comment

Experimental design

No comment

Validity of the findings

No comment

Additional comments

The revised manuscript can be accepted.